# Multi-View Graph Clustering via Node-Guided Contrastive Encoding

Yazhou Ren [1 2]  JunLong Ke [1]  Zichen Wen [1]  Tianyi Wu [1]  Yang Yang [1]  Xiaorong Pu [1 2]  Lifang He [3]

## Abstract

Multi-view clustering has gained significant attention for integrating multi-view information in multimedia applications. With the growing complexity of graph data, multi-view graph clustering (MVGC) has become increasingly important. Existing methods primarily use Graph Neural Networks (GNNs) to encode structural and feature information, but applying GNNs within contrastive learning poses specific challenges, such as integrating graph data with node features and handling both homophilic and heterophilic graphs. To address these challenges, this paper introduces Node-Guided Contrastive Encoding (NGCE), a novel MVGC approach that leverages node features to guide embedding generation. NGCE enhances compatibility with GNN filtering, effectively integrates homophilic and heterophilic information, and strengthens contrastive learning across views. Extensive experiments demonstrate its robust performance on six homophilic and heterophilic multi-view benchmark datasets.

## 1. Introduction

Multi-view clustering (MVC) is an important unsupervised learning task in the machine learning community (Li et al., 2014; Xu et al., 2022; Yin et al., 2020; Tan et al., 2023; Zhang et al., 2023; Liu et al., 2020; Chao et al., 2024; Guo & Ye, 2019; Zhang et al., 2024; Liu et al., 2019; Wu et al., 2020). It is designed to integrate diverse and complementary information across multiple views for clustering(Ke et al., 2024; Huang et al., 2021; Ren et al., 2024a; Wu et al., 2024; Ren et al., 2024b). Recent advances in unsupervised learning for graph data have increasingly emphasized the

---

[1]School of Computer Science and Engineering, University of Electronic Science and Technology of China [2]Shenzhen Institute for Advanced Study, University of Electronic Science and Technology of China [3]Department of Computer Science and Engineering, Lehigh University. Correspondence to: Junlong Ke <junlong.ke@outlook.com>.

*Proceedings of the 42nd International Conference on Machine Learning*, Vancouver, Canada. PMLR 267, 2025. Copyright 2025 by the author(s).

distinction between the data associated with the ego node and that of its edges, which delineate relationships with neighboring nodes (Tang et al., 2022; Xiao et al., 2022; Chen et al., 2022). However, some methods overly emphasize edge and neighborhood data, often neglecting the intrinsic information of the ego node. This imbalance has led to suboptimal performance on certain datasets, sometimes even underperforming compared to Multi-Layer Perceptrons (MLPs) (Tang et al., 2022; Xiao et al., 2022). On the other hand, some approaches rely heavily on the ego node's attributes to maintain model performance after multiple layers of processing in Graph Neural Networks (GNNs). While this preserves some robustness, it limits the ability to leverage contextual information from neighboring nodes, leading to subpar results on specific datasets (Chen et al., 2022).

In the realm of graph contrastive learning for multi-view graph clustering (MVGC), capturing node differences and diversity within graphs plays a crucial role in designing effective contrastive learning methodologies (Hassani & Khasahmadi, 2020). Traditional approaches have largely relied on the homophily assumption (McPherson et al., 2001), but more recent studies have begun to explore the effectiveness of these methods in the context of heterophilic graphs. For example, COMPLETER (Contrastive Prediction for Incomplete Multi-view Clustering) (Lin et al., 2021) has been introduced for heterophilic graphs. It integrates reconstruction, cross-view contrastive learning, and cross-view dual prediction to simultaneously achieve data recovery and consistency learning for incomplete multi-view data. (Hassani & Khasahmadi, 2020) proposes contrastive multi-view representation learning on graphs (MVGRL) by performing representation learning by contrasting two diffusion matrices transformed from the adjacency matrix. HGRL (Chen et al., 2022) leverages the preservation of original node features and the capturing of non-local neighbors to enhance node representations on heterophilic graphs. GREET (Liu et al., 2023) distinguishes between homophilic and heterophilic edges, employing low-pass and high-pass filters to seize the corresponding information. Building upon generation methods, DSSL (Xiao et al., 2022) decouples diverse patterns in local neighborhood distribution to capture both homophilic and heterophilic information. NWR-GAE (Tang et al., 2022) underscores the significance of topological structure

within graphs, reconstructing neighborhoods based on local structure and features. MUSE (Yuan et al., 2023) innovates by creating two subviews on the original view, applying an information fusion controller for view utilization.

However, these methodologies either lack efficacy in heterophilic graphs or underperform in homophilic scenarios. We attribute this limitation to an explicit differentiation between homophilic and heterophilic information through decoupling or view reconstruction, or an overemphasis on one aspect over the other. This leads to an inadequate interaction between homophilic and heterophilic information during contrastive analysis. A more natural approach would be to integrate both aspects within a unified representation framework, raising the key question: ***How can we encode both types of information without compromising either?***

Our approach involves guiding view feature encoding with node features, followed by adaptive weighting that accounts for graph homophily and joint information aggregation using a similarity matrix derived from refined node embeddings. Recognizing the strengths of traditional GNNs in graph structure mining, as well as the challenges posed by GNN transformations, we propose encoding the original graph structure with GNNs while integrating node features to adapt representations for traditional GNNs. This approach eliminates the need to explicitly distinguish between homophilic and heterophilic graph components. We begin by constructing a node correlation matrix and employing pseudo-labeling to quantify the homophily of the original adjacency matrix, thereby allocating weights to selectively preserve original structural information. Additionally, we introduce a random mask to the original node feature information, generating an additively noisy feature matrix. Subsequently, we leverage GNNs' message passing and neighborhood aggregation mechanisms to recover the noisy features, with a contrastive loss applied between the recovered and original feature representations. Finally, we utilize inter-view contrastive learning to foster complementary learning across views. The main contributions of this paper can be summarized as follows.

- We propose NGCE, a novel contrastive learning model for homophilic and heterophilic graphs. To the best of our knowledge, NGCE is the first graph contrastive learning method guided by the principle that homophilic and heterophilic information in graph data should not be isolated, but rather processed within a unified node-guided framework to preserve its interactive essence in MVGC.

- We develop an effective joint process for MVGC that integrates several key components: an edge and node embedding similarity matrix sensitive to graph homophily, a contrastive learning-guided graph encoding mechanism driven by the recovery of noise-enhanced

node features, and a contrastive fusion mechanism across views. Together, these elements form a comprehensive process for node-guided multi-view co-coding.

- We conduct extensive experiments on six homophilic and heterophilic benchmark datasets to evaluate the performance of NGCE. Our results show that the proposed NGCE adeptly accommodates both homophilic and heterophilic datasets within the multi-view graph clustering domain, achieving state-of-the-art performance metrics.

## 2. Related Work

### 2.1. Multi-View Graph Clustering

With the rapid progress of GNNs, there has been a growing interest among researchers in leveraging graph structural information for multi-view clustering (MVGC). By combining with graph convolutional encoders, MVGC aims to learn the graph's underlying semantic information and divide the nodes into different clusters. In recent years, a plethora of methods addressing MVGC have surfaced. O2MAC (Fan et al., 2020) can be credited as a pioneer in applying GNNs to MVGC. Their approach involves encoding multi-view graphs into a lower-dimensional space using a single-view graph convolutional encoder and a multi-view graph structure decoder. Building on this foundation, (Cheng et al., 2020) introduces a novel design of two-pathway graph encoders, facilitating the mapping of graph embedding features and the acquisition of view-consistent information. Meanwhile, (Hassani & Khasahmadi, 2020) proposes an innovative GNN-based solution specifically tailored for multiview self-supervised learning, aiming to acquire both node and graph-level representations. In a similar vein, (Pan & Kang, 2021) applies contrastive learning techniques to unearth shared geometric and semantic patterns, enhancing the learning of a consensus graph. (Xia et al., 2022b) takes a systematic approach, exploring cluster structures through a graph convolutional encoder trained to learn the self-expression coefficient matrix. However, a common challenge across these methods lies in their high sensitivity to the homophily of graphs. In other words, when these models are applied to graphs exhibiting strong heterophily, their effectiveness may fall short of expectations.

### 2.2. Contrastive Learning

Contrastive learning has garnered considerable attention in the realm of unsupervised learning, showcasing remarkable performance across a multitude of tasks, particularly on images (Hjelm et al., 2018; Chen et al., 2020b; Grill et al., 2020; Zbontar et al., 2021; Zhong et al., 2021; Yang et al., 2024) and graphs (Velickovic et al., 2019; You et al., 2020; Zhu et al., 2020; Xia et al., 2022a; Bielak et al., 2022; Hu et al., 2021). The underlying principle involves maximizing

the similarity among positive pairs while minimizing the distance between negative pairs, as outlined by (Hadsell et al., 2006). In general, positive pairs consist of augmented versions of the same instance, while pairs involving different instances are considered negatives. Various loss functions have been introduced for this purpose, including the triplet loss (Chopra et al., 2005), noise contrastive estimation (NCE) loss (Gutmann & Hyvärinen, 2010), and normalized temperature-scaled cross-entropy loss (NT-Xent) (Chen et al., 2020a). Presently, numerous research endeavors leverage contrastive learning methods in the realm of clustering tasks, significantly contributing to the progress of clustering and unsupervised representation learning. (Zhong et al., 2020) suggests converting the maximization of mutual information into minimizing contrast loss, resulting in notable enhancements when applying contrast learning to diminish within-class variability. However, it fails to accommodate the prevalent graph data. (Hassani & Khasahmadi, 2020) proposes contrastive multi-view representation learning on graphs (MVGRL) by performing representation learning by contrasting two diffusion matrices transformed from the adjacency matrix. However, it relies heavily on data augmentation techniques and primarily utilizes graph structure information, specifically the adjacency matrix. We leverage node feature information to guide graph-contrastive encoding, thereby obtaining enhanced graph representations, harmonizing homophilic and heterophilic information in MVGC, and preserving their interactive essence.

## 3. Method

As shown in Fig. 1, the proposed approach consists of two components: i) **Node-Guided Contrastive Encoding Module**: This module introduces an innovative encoding strategy that achieves graph joint encoding across different views, guided by the node's original features. This approach comprises two key modules: an adaptive weighted sum based on similarity graphs and contrastive learning, implemented after reconstructing node features through GNN under the influence of additive noise. The former is tailored to integrate node features into the joint encoding output adaptively, whereas the latter aims to guide the encoding process by enabling the joint encoding to exhibit restorative properties that are based on the noise-augmented original node features. This process is designed to ensure that the integrity of homophilic and heterophilic information exchange within the graph data is preserved, thereby yielding a universally applicable output encoding; ii) **Cross-View Contrast Module**: To enhance the learning of view consistency and complementarity, we adapt the centroids of clusters and the $k$-nearest neighbor samples as additional positive and negative samples, respectively, significantly improving the efficacy of view integration.

### 3.1. Node-Guided Contrastive Encoding Module

#### 3.1.1. GRAPH JOINT PROCESS

To enhance the proposed framework's capability for effective neighborhood integration, it is critical to ensure that a substantial portion of nodes adjacent to the encoded embeddings share the same class as the node in focus since the original node features will be unavailable for later processing. Considering that nodes within the same category have comparable feature vectors due to identical labels, our methodology prioritizes exploiting this feature similarity for joint graph encoding that combines edge and node relationships.

Initially, we employ autoencoders to distill and refine the feature vectors of nodes within the graph:

$$\mathbf{Z}^n = f^n(\sigma(\mathbf{X}; \theta^n)), \tag{1}$$

$$\mathbf{X}^n_{\mathbf{pred}} = g^n(\sigma(\mathbf{Z}^n; \varphi^n)), \tag{2}$$

where, for the $n$-th view, $\mathbf{Z}^n$ refers to the node embeddings, $\theta^n$ and $\varphi^n$ refers to the encoder and decoder's learnable parameters that are unshared, respectively, $\mathbf{X}$ refers to the original node features, while $\sigma(\cdot)$ refers to the activation function.

Subsequently, to uncover node correlations, we evaluate the cosine similarity between node features, leading to the formulation of the correlation matrix $\mathbf{S}^n$ for the $n$-th view:

$$\mathbf{S}^n = \text{Sim}(\mathbf{Z}^n, \mathbf{Z}^{n\text{T}}), \tag{3}$$

where the function $\text{Sim}(\cdot, \cdot)$ refers to cosine similarity in vector space.

Using $\mathbf{S}^n$ as the basis for encoding the graph does improve its homophily to some extent. However, focusing solely on node features while ignoring the inherent structural aspects of the graph may not optimally serve the encoding goals. Therefore, we argue for a strategic inclusion of the structural nuances of the original graph in our encoding efforts to ensure an increased level of homophily. Considering that the higher homophily degree indicates that the similarity information on the graph is dominant, and vice versa for the dissimilarity information, this involves the development of an adaptive graph encoding strategy where we aim to evaluate and weigh the degree of homophily of the original graph. Naturally, we utilize the homophily degree to assign weights to $\widehat{\mathbf{A}}^n$ and $\mathbf{S}^n$. However, the homophily degree of a given graph is difficult to measure due to the lack of true labels in the unsupervised scenario. Thus, we propose to learn the pseudo-label information (Arazo et al., 2020) using the consensus embedding $\mathbf{H}$ of Eq. (12) of the last iteration, and use the pseudo-label information and adjacency relationship information to approximate the homophily weight $\omega^n$ for

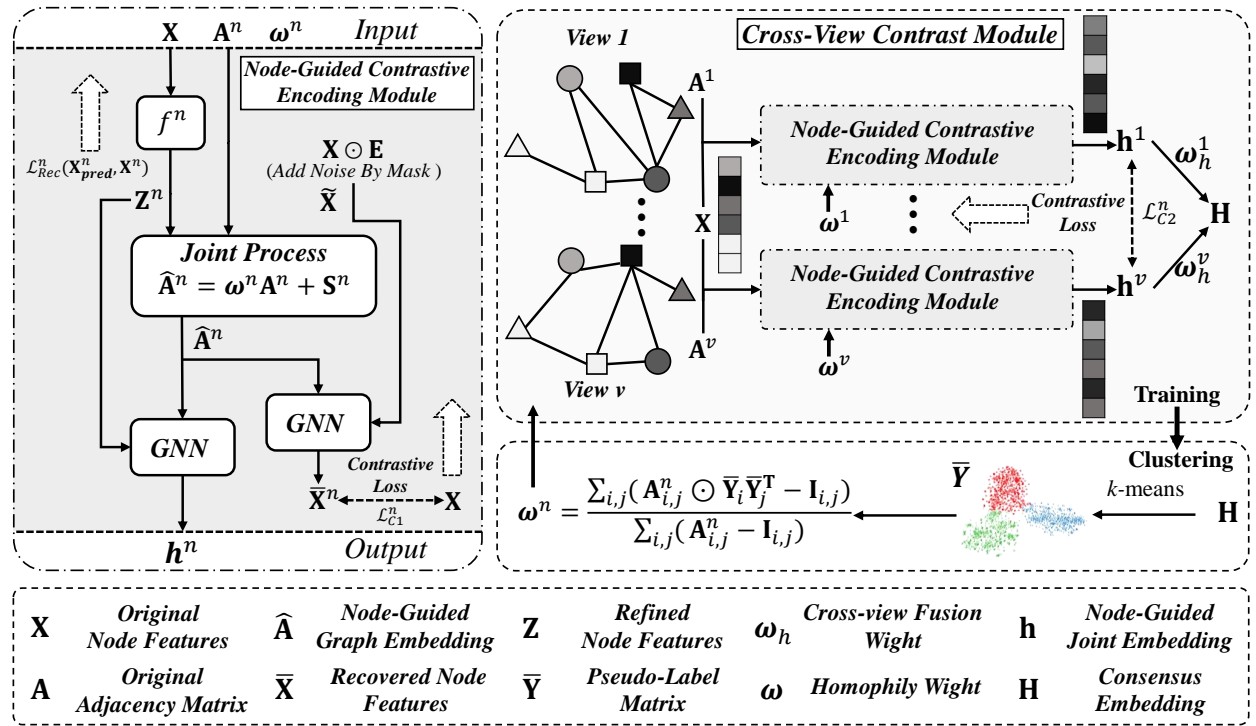

| | | | | | | | |
|---|---|---|---|---|---|---|---|
| $\mathbf{X}$ | *Original Node Features* | $\widehat{\mathbf{A}}$ | *Node-Guided Graph Embedding* | $\mathbf{Z}$ | *Refined Node Features* | $\omega_h$ | *Cross-view Fusion Wight* | $\mathbf{h}$ | *Node-Guided Joint Embedding* |
| $\mathbf{A}$ | *Original Adjacency Matrix* | $\overline{\mathbf{X}}$ | *Recovered Node Features* | $\overline{\mathbf{Y}}$ | *Pseudo-Label Matrix* | $\omega$ | *Homophily Wight* | $\mathbf{H}$ | *Consensus Embedding* |

*Figure 1.* The illustration of NGCE graph clustering framework. The inputs to each view are the node feature matrix $\mathbf{X}$ and the adjacency matrix $\mathbf{A}$ of views. The output of the framework is the consensus embedding $\mathbf{H}$, after which $\mathbf{H}$ is used as input for $K$-means clustering. Dashed arrows represent the backward propagation.

the $n$-th view:

$$\omega^n = \frac{\sum_{i,j}(\mathbf{A}_{i,j}^n \odot \overline{\mathbf{Y}}_i \overline{\mathbf{Y}}_j^T - \mathbf{I}_{i,j})}{\sum_{i,j}(\mathbf{A}_{i,j}^n - \mathbf{I}_{i,j})}, \qquad (4)$$

where, $\omega^n$ refers to the weight reflective of the homophily degree from the preceding iteration, with $\odot$ indicating the Hadamard product. $\overline{\mathbf{Y}} \in \{0,1\}^{N \times c}$ representing the one-hot encoded pseudo labels from the clustering of the consensus embedding $\mathbf{H}$, where $N$ refers to the number of nodes and $c$ refers to the number of clusters. $\mathbf{I}$ refers to the identity matrix.

Ultimately, this leads to graph joint encoded embeddings that integrate both the homophily assessment and the original structural framework:

$$\widehat{\mathbf{A}}^n = \mathbf{S}^n + \omega^n \mathbf{A}^n, \qquad (5)$$

In order to guide the autoencoder $f^n(\cdot)$, we apply reconstruction loss on $\mathbf{X}_{\mathbf{pred}}^n$ and $\mathbf{X}$ as follows:

$$\begin{aligned} \mathcal{L}_{Rec}^n &= l(g^n(\mathbf{X}_{\mathbf{pred}}^n; \mathbf{X}) \\ &= l(g^n(\sigma(f^n(\sigma(\mathbf{X}; \theta^n)); \varphi^n)); \mathbf{X}), \end{aligned} \qquad (6)$$

where $l(\cdot; \cdot)$ denotes loss function.

### 3.1.2. NODE-GUIDED CONTRASTIVE ENCODING

Furthermore, we design the node-guided optimization process, aiming to harness the true features of nodes to steer the encoding phase within the joint process. Inspired by the Denoising Autoencoder (DAE) (Vincent et al., 2008) and GraphMAE (Hou et al., 2022), we replaced the encoder and decoder in DAE with parameter-free GNNs. In this adaptation, node features are treated as denoising coding objects, which facilitates the training of the node-guided graph embedding $\widehat{\mathbf{A}}$. In this process, $\widetilde{\mathbf{A}}$ works analogous to the parameters of the encoder and decoder.

More precisely, we actively encourage the post-encoding aggregation procedure, guided by the encoded graph, to exhibit restorative capabilities towards the original features after noise is masked. This strategy, as substantiated by subsequent experimental validations, proves to be compatible with both homophilic and heterophilic graph data. Through such a design, we not only maintain the interaction between homophilic and heterophilic information within the graph data but also guide an effective excavation of this information. This methodology underscores the importance of preserving the intrinsic data structure while facilitating a nuanced representation of the graph data, accommodating both similarity and diversity within the graph structure.

We first add a random mask as noise to the original matrix $\mathbf{X}$ of node features. In practice, we randomly drop some node features in the graph with a probability of $p_c$ by adding a random mask as noise to the original matrix $\mathbf{X}$. Specifically, we sample a binary masking matrix $\mathbf{E} \in \{0,1\}^{N \times d}$ from the Bernoulli distribution with a probability of $(1 - p_c)$, *i.e.*, $e_{ij} \sim Bernoulli\,(1 - p_c), i, j \in \{1, \cdots, N\}$, and perform element-wise multiplication with the node features matrix:

$$\widetilde{\mathbf{X}} = \mathbf{X} \odot \mathbf{E}, \tag{7}$$

where $\odot$ denotes the Hadamard product, $\mathbf{X}$ represents the original node features and $\widetilde{\mathbf{X}}$ refers to the node features with random mask (noisy features).

Then, we utilize GCN's neighborhood aggregation mechanism to recover the feature information following the addition of noise, which can be represented from $\mathbf{X} \to \widetilde{\mathbf{X}} \to \overline{\mathbf{X}}^n$, where $\overline{\mathbf{X}}^n = GCN(\widehat{\mathbf{A}}^n, \widetilde{\mathbf{X}})$ refers to the node features recovered by the aggregation mechanism of parameter-free GCN: $\overline{\mathbf{X}}^n = GCN(\widehat{\mathbf{A}}^n, \widetilde{\mathbf{X}})$, which can be formulated as:

$$\mathbf{H}^{(0)} = \widetilde{\mathbf{X}},$$
$$\mathbf{H}^{(l+1)} = \sigma\left(\mathbf{D}^{-\frac{1}{2}}\widehat{\mathbf{A}}\mathbf{D}^{-\frac{1}{2}}\mathbf{H}^{(l)}\right) + \widetilde{\mathbf{X}}, \tag{8}$$

where $\mathbf{H}^{(l)}$ refers to the output of $l$-th layer while $\mathbf{D}$ refers to the degree matrix of $\widehat{\mathbf{A}}$. Residual connections have been integrated to avoid potential gradient problems. The depth of our GCN is defined in terms of $order$, which is set as a fixed hyperparameter in our framework. We force the $n$-th view's recovered node features matrix $\overline{\mathbf{X}}^n$ to be equal to the original node features $\mathbf{X}$ as formulated:

$$\begin{aligned}\mathcal{L}_{C1}^n &= \frac{1}{N^2}\sum(\overline{\mathbf{X}}^n - \mathbf{X})^2 = \frac{1}{N^2}\sum_i\sum_j(\overline{\mathbf{X}}_{ij}^n - \mathbf{X}_{ij})^2 \\ &= \frac{1}{N^2}\sum_i\sum_j((\overline{\mathbf{X}}_{ij}^n - \widetilde{\mathbf{X}}_{ij}) - (\mathbf{X}_{ij} - \widetilde{\mathbf{X}}_{ij}))^2 \\ &= \frac{1}{N^2}(\sum_i\sum_j\mathbb{I}_{ij}^1(\overline{\mathbf{X}}_{ij}^n - \widetilde{\mathbf{X}}_{ij})^2 \\ &\quad + \sum_i\sum_j\mathbb{I}_{ij}^0(\overline{\mathbf{X}}_{ij}^n - \mathbf{X}_{ij})^2),\end{aligned} \tag{9}$$

where, the $\mathbb{I}_{ij}^0$ and $\mathbb{I}_{ij}^1$ functions work as indicators, with the former being 1 when $\mathbf{E}_{ij} = 1$ and 0 otherwise, and the latter vice versa. Specifically, in Eq. (9), the first term minimizes the agreement between the reconstructed node features and the features of the masked nodes in the encoder graph, since there is no valid information correlation in these nodes; the second term drives the GCN with graph joint encoding embeddings to retain critical unmasked node information.

In other words, given the variation component as $\overline{\mathbf{X}}_{ij}^n - \widetilde{\mathbf{X}}_{ij}$,

we can reformulate the loss function such that:

$$\begin{aligned}\mathcal{L}_{C1}^n &= \frac{1}{N^2}(\sum_i\sum_j\mathbb{I}_{ij}^1(\overline{\mathbf{X}}_{ij}^n - \widetilde{\mathbf{X}}_{ij})^2 \\ &\quad + \sum_i\sum_j\mathbb{I}_{ij}^0((\overline{\mathbf{X}}_{ij}^n - \widetilde{\mathbf{X}}_{ij}) - \mathbf{X}_{ij})^2),\end{aligned} \tag{10}$$

where the first term minimizes the change in features when node features are preserved, while the second term forces nodes to conform to their original features when node features are not preserved, essentially establishing an adaptive mechanism for selecting positive and negative samples. This involves selecting unpreserved features as positive samples and the variation component of preserved features themselves as negative samples.

For the $n$-th view, considering the dimensionality of node-guided graph embedding $\widehat{\mathbf{A}}^n$ is excessively large for cross-view fusion processes, we employ parameter-free GCNs once as a dimensionality reduction mechanism. This process results in compressed representations $\mathbf{h}^n$ that are dimensionally aligned with node features $\mathbf{Z^n}$:

$$\mathbf{h}^n = GCN(\widehat{\mathbf{A}}^n, \mathbf{Z}^n), \tag{11}$$

we obtain node-guided joint embeddings $\mathbf{h}^n$ for the $n$-th view.

### 3.2. Cross-View Contrast Module

In the context of multi-view tasks, it is acknowledged that distinct views offer varying levels of information, characterized by both consistency and complementarity (Jia et al., 2020; Wu et al., 2019). Initially, we obtain node-guided embeddings $\mathbf{h}^n$ for $n$-th view. To optimally leverage this diverse yet complementary information across views, our approach aims to synthesize a consensus embedding rich in information by amalgamating the node-guided joint embeddings $\mathbf{h}^n$ from each individual view (Jia et al., 2022). Acknowledging the differential informational value across views necessitates the implementation of a weighting mechanism, predicated on an assessment of each view's informational quality. This approach is designed to ensure that contributions from various views to the overarching consensus embedding are proportionate to their evaluated significance. In essence, a view's embedding that exhibits a high degree of similarity to the consensus embedding $\mathbf{H}$ is deemed to possess significant information, thereby warranting a higher weight allocation, and the inverse applies for lesser similarity. The consensus embedding $\mathbf{H}$ is derived as follows:

$$\mathbf{H} = \sum_{n=1}^{V}\boldsymbol{\omega}_h^n\mathbf{h}^n, \tag{12}$$

where $\boldsymbol{\omega}_h^n$ represents the weight allocated to the node-guided

joint embeddings of the $n$-th view, determined through:

$$\boldsymbol{\omega}_h^n = \left(\frac{eva^n}{\max\left(eva^1, eva^2, \cdots, eva^V\right)}\right)^\rho. \quad (13)$$

This weight, $\boldsymbol{\omega}_h^n$, is deduced from an evaluation function designed to measure the congruence between the consensus embedding $\mathbf{H}$ and each view's embedding $\mathbf{h}^n$, *i.e.*, $eva^n = evaluation(\mathbf{h}^n, \mathbf{H})$. The hyperparameter $\rho$ facilitates the modulation of view weights, either enhancing or diminishing their impact. To culminate, the consensus embedding $\mathbf{H}$ undergoes $k$-means clustering to ascertain the clustering outcomes.

Inspired by the methodologies outlined in (Yuan et al., 2023; Hassani & Khasahmadi, 2020), we refine the joint node representation $\mathbf{H}$ by integrating a contrastive loss component. This component aims to further enhance the learning of view complementarity reflected in $\mathbf{H}$. We employ a variant of NT-Xent loss (Chen et al., 2020a) as our contrastive learning loss function. Crucially, for an embedding $\mathbf{h}_i$ of node $i$ within the consensus embedding $\mathbf{H}$, we select the embeddings of its $N_{knn}$ nearest neighbor nodes within $\mathbf{H}$ as positive samples. For the selection of negative samples, we adopt a method inspired by that described in (Chao et al., 2024). Specifically, after a predetermined number of training epochs, once the cluster centers are considered sufficiently stable to represent high-confidence and disparate class prototypes, we opt for representations from cluster centers of different classes ($C_k$ where $k \neq c_i$), which are identified from $\mathbf{H}$ in the previous iteration and represent high-confidence, disparate class prototypes. This process can be formulated as follows:

$$\mathcal{L}_{C2} = -\mathcal{A}\mathbb{1}_{[t \geq T_1]} \sum_{i=1}^{N} \log\left(\sum_{m=1}^{N_{knn}} e^{\mathcal{S}(\mathbf{h}_i, \mathbf{p}_{i,m})} \middle/ \right.$$
$$\left. \left(\sum_{m=1}^{N_{knn}} e^{\mathcal{S}(\mathbf{h}_i, \mathbf{p}_{i,m})} + \sum_{k=1}^{c} \mathbb{1}_{[k \neq c_i]} e^{\mathcal{S}(\mathbf{h}_i, C_k)}\right)\right), \quad (14)$$

where $\mathbf{h}_i$ refers to the feature of node $i$ within the consensus embedding $\mathbf{H}$. $\mathbf{p}_{i,m}$ refers to the feature representation of the $m$-th nearest neighbor of node $i$, drawn from $P(\mathbf{h}_i)$, which denotes the set of $N_{knn}$ nearest neighbors of $\mathbf{h}_i$ in $\mathbf{H}$. $N_{knn}$ indicates the number of nearest neighbors selected, which is fixed at 20 in our implementation. $\mathcal{S}(\cdot, \cdot)$ denotes the cosine similarity function. $N$ refers to the total number of nodes, while $c$ represents the total number of clusters. $\mathcal{A}$ denotes the loss coefficient, and we have empirically set its value to $\frac{1}{N(c-1)}$ based on experimental results. The indicator function $\mathbb{1}_{[\cdot]}$ yields a value of 1 if the condition specified in $[\cdot]$ is met, and 0 otherwise. Herein, $t$ denotes the current training epoch and $T_1$ is a predefined hyperparameter; this loss component is thereby activated only when $t \geq T_1$. $C_k$ represents the cluster center for the $k$-th class, and $c_i$

| Datasets | Clusters | Nodes | Features | Graphs $(hr)$ |
|---|---|---|---|---|
| ACM | 3 | 3025 | 1870 | $\mathcal{G}^1(0.82), \mathcal{G}^2(0.64)$ |
| DBLP | 4 | 4057 | 334 | $\mathcal{G}^1(0.80), \mathcal{G}^2(0.67), \mathcal{G}^3(0.32)$ |
| IMDB | 3 | 4780 | 1232 | $\mathcal{G}^1(0.48), \mathcal{G}^2(0.62), \mathcal{G}^3(0.40)$ |
| Texas | 5 | 183 | 1703 | $\mathcal{G}^1(0.09), \mathcal{G}^2(0.09)$ |
| Chameleon | 5 | 2277 | 2325 | $\mathcal{G}^1(0.23), \mathcal{G}^2(0.23)$ |
| Wisconsin | 5 | 251 | 1703 | $\mathcal{G}^1(0.19), \mathcal{G}^2(0.19)$ |

*Table 1.* The statistics information of the six graph datasets. $(hr)$ refers to the homophily rate calculated by true labels.

indicates the class assigned to node $i$. Both $C_k$ and $c_i$ are derived from the $k$-means clustering result applied to $\mathbf{H}$ from a previous iteration.

### 3.3. Model Optimization

Learning from previous deep clustering studies (Ren et al., 2025), we design the overall optimization objective for NGCE, which can be formulated as:

$$\mathcal{L}_{total} = \mathcal{L}_C + \mathcal{L}_{Rec}$$
$$= \sum_{n=1}^{V} \mathcal{L}_{C1}^n + \mathcal{L}_{C2}^n + \mathcal{L}_{Rec}^n. \quad (15)$$

$\mathcal{L}_{total}$ can be divided into two main sections: unsupervised node embedding learning and contrastive learning. The unsupervised node embedding learning component $\mathcal{L}_{Rec}$ drives the optimization of refined node features $\mathbf{Z}$. The contrastive learning component $\mathcal{L}_C$ includes the loss functions $\mathcal{L}_{C1}$ and $\mathcal{L}_{C2}$. In the former, we apply noisy masks as perturbations to node features to drive the optimization of node-guided graph embedding $\widehat{\mathbf{A}}$, while in the latter, we utilize contrastive learning based on cluster centroids to optimize cross-view consensus embedding $\mathbf{H}$. For the sake of cross-dataset robustness and simplicity in hyperparameter tuning, we omitted the weight of the loss term.

## 4. Experiments

### 4.1. Experimental Setup

#### 4.1.1. DATASETS.

We used six MVG datasets. The homophilous graph datasets include: **ACM** (Fan et al., 2020), **DBLP** (Fan et al., 2020) and **IMDB** (Fan et al., 2020). The heterophilous graph datasets include: **Texas**, **Chameleon** (Rozemberczki et al., 2021) and **Wisconsin** (Pei et al., 2020). In the appendix, we provide details and sources of the datasets.

Table 1 summarizes the statistics of these six datasets. The top half of the table shows the statistics information for the homophilous graph datasets and the bottom half shows the

| Methods | ACM (hr 0.82 & 0.64) | | | | DBLP (hr 0.80 & 0.67 & 0.32) | | | | IMDB (hr 0.48 & 0.62 & 0.40) | | | |
|---|---|---|---|---|---|---|---|---|---|---|---|---|
| | NMI% | ARI% | ACC% | F1% | NMI% | ARI% | ACC% | F1% | NMI% | ARI% | ACC% | F1% |
| VGAE (2016) | 49.1 | 54.4 | 82.2 | 82.3 | 69.3 | 74.1 | 88.6 | 87.4 | 0.4 | 0.9 | 44.2 | 35.7 |
| DAEGC (2019) | 63.8 | 70.1 | 89.0 | 88.9 | 30.8 | 33.4 | 66.5 | 65.6 | 0.6 | 1.0 | 37.9 | 35.3 |
| AGE (2020) | 73.5 | 78.9 | 92.4 | 92.4 | 45.0 | 47.6 | 75.3 | 74.6 | 4.4 | 4.6 | 43.2 | 42.2 |
| O2MAC (2020) | 69.2 | 73.9 | 90.4 | 90.5 | 72.9 | 77.8 | 90.7 | 90.1 | 0.3 | 0.2 | 40.2 | 35.4 |
| MvAGC (2020) | 67.4 | 72.1 | 89.8 | 89.9 | 77.2 | 82.8 | 92.8 | 92.3 | 1.3 | -1.8 | 48.5 | 28.2 |
| AGCN (2021) | 68.4 | 74.2 | 90.6 | 90.6 | 39.7 | 42.5 | 73.3 | 72.8 | 0.3 | 1.4 | 54.5 | 31.1 |
| MCGC (2021) | 71.3 | 76.3 | 91.5 | 91.6 | **83.0** | 77.5 | 93.0 | 92.5 | 5.2 | 10.3 | **58.3** | 38.8 |
| DCRN (2022) | 71.6 | 77.6 | 91.9 | 91.9 | 49.0 | 53.6 | 79.7 | 79.3 | 0.2 | 0.1 | 53.4 | 25.5 |
| DuaLGR (2023) | 73.2 | 79.4 | 92.7 | 92.7 | 75.5 | 81.7 | 92.4 | 91.8 | _6.2_ | _12.5_ | 52.0 | _44.7_ |
| CMGEC (2023) | 69.1 | 72.3 | 90.9 | 90.7 | 72.4 | 78.6 | 91.0 | 90.4 | 5.1 | 4.7 | 48.4 | **51.0** |
| BMGC (2024) | _78.4_ | _83.3_ | _94.1_ | _94.2_ | _80.1_ | **85.4** | **94.0** | **93.6** | 5.5 | 4.9 | 44.1 | 40.7 |
| SMVC (2024) | 72.4 | 78.0 | 92.3 | 92.0 | 76.1 | 81.6 | 92.4 | 92.0 | **8.0** | 7.2 | 41.3 | 37.2 |
| VGMGC (2025) | 76.3 | 81.9 | 93.6 | 93.6 | 78.3 | 83.7 | 93.2 | 92.7 | 0.8 | 3.2 | 52.6 | 32.8 |
| NGCE (ours) | **80.5** | **85.0** | **94.7** | **94.8** | 79.1 | _84.0_ | _93.3_ | _92.8_ | 5.6 | **12.7** | _54.6_ | 43.4 |

*Table 2.* The results of clustering on homophilous graph datasets. We express all evaluative metrics as percentages. The best and runner-up results are highlighted with **bold** and underline, respectively.

| Methods | Texas (hr 0.09 & 0.09) | | | | Chameleon (hr 0.23 & 0.23) | | | | Wisconsin (hr 0.19 & 0.19) | | | |
|---|---|---|---|---|---|---|---|---|---|---|---|---|
| | NMI% | ARI% | ACC% | F1% | NMI% | ARI% | ACC% | F1% | NMI% | ARI% | ACC% | F1% |
| VGAE (2016) | 12.7 | 21.7 | 55.3 | 29.5 | 15.1 | 12.4 | 35.4 | 29.6 | 10.5 | 13.7 | 49.3 | 34.1 |
| DAEGC (2019) | 6.4 | 2.6 | 31.7 | 25.0 | 9.1 | 5.6 | 32.2 | 31.2 | 10.6 | 3.4 | 32.7 | 28.3 |
| AGE (2020) | 7.5 | 7.3 | 36.6 | 36.6 | 8.6 | 7.6 | 32.4 | 32.4 | 9.3 | 1.3 | 31.1 | 31.1 |
| O2MAC (2020) | 8.7 | 14.6 | 46.7 | 29.1 | 12.3 | 8.9 | 33.5 | 28.6 | 11.0 | 8.9 | 40.0 | 27.9 |
| MvAGC (2020) | 5.4 | 1.1 | 54.3 | 19.8 | 10.8 | 3.3 | 29.2 | 24.3 | 8.1 | 4.8 | 47.7 | 20.6 |
| AGCN (2021) | 15.4 | 18.1 | _61.8_ | 43.0 | 6.7 | 6.1 | 32.5 | 20.4 | 6.4 | 6.8 | 49.8 | 24.9 |
| MCGC (2021) | 12.7 | 12.9 | 51.9 | 32.5 | 9.5 | 5.9 | 30.0 | 19.1 | 12.9 | 5.9 | 51.8 | 30.7 |
| DCRN (2022) | 10.7 | 15.1 | 55.2 | 27.6 | 8.7 | 5.7 | 30.9 | 21.9 | 10.8 | 16.0 | 50.2 | 34.1 |
| DuaLGR (2023) | 32.6 | _26.0_ | 54.3 | _46.4_ | 19.5 | _16.0_ | _41.1_ | 37.7 | 34.1 | 28.8 | 56.4 | 47.1 |
| BMGC (2024) | 29.1 | 15.8 | 42.5 | 38.3 | 9.4 | 5.9 | 30.8 | 30.7 | 34.0 | 24.3 | 51.5 | 40.8 |
| VGMGC (2025) | _35.4_ | _26.0_ | 55.2 | **46.9** | _22.4_ | 13.4 | 40.1 | **39.5** | _41.6_ | _34.8_ | _56.6_ | **49.6** |
| NGCE (ours) | **47.8** | **54.9** | **77.6** | 46.2 | **22.6** | **19.3** | **42.2** | _38.4_ | **46.8** | **46.4** | **73.7** | _47.2_ |

*Table 3.* The results of clustering on heterophilous graph datasets. We express all evaluative metrics as percentages. The best and runner-up results are highlighted in **bold** and underlined, respectively.

statistics information for the heterophilous graph datasets.

#### 4.1.2. BASELINES.

Several baselines are replicated for comparison with our model. VGAE (Kipf & Welling, 2016) is a classical single-view clustering method. O2MAC (Fan et al., 2020) is the method that learns from both node features and graphs. MvAGC (Lin & Kang, 2021) and MCGC (Pan & Kang, 2021) are two methods based on graph filters to learn a consensus graph for clustering. DualGR (Ling et al., 2023) leverages soft-label and pseudo-label to guide the graph refinement and fusion process for clustering. CMGEC (Wang et al., 2023) utilizes a multi-graph attention fusion encoder with multi-view mutual information maximization mod-

ule, aiming to encode more complementary information from multiple views and depict data more comprehensively. BMGC (Shen et al., 2024) addresses view imbalance in multi-relational graphs by proposing unsupervised dominant view mining and dual signals guided representation learning. SMVC (Chen et al., 2024) is a structural deep multi-view clustering method that integrates top-level abstraction with underlying details to jointly optimize cluster assignments and feature embeddings. VGMGC (Chen et al., 2025) introduces a variational graph generator to infer a reliable consensus graph from multiple graphs, aiming to better utilize both view-specific and view-common information for multiview graph clustering. For all baselines, we use results reported in the original papers or conduct experiments with their default parameters setting.

| Methods | ACM ($hr$ 0.82 & 0.64) | | | | Chameleon ($hr$ 0.23 & 0.23) | | | |
|---|---|---|---|---|---|---|---|---|
| | NMI% | ARI% | ACC% | F1% | NMI% | ARI% | ACC% | F1% |
| NGCE (w/o $\mathcal{L}_{Rec}$) | 62.4 | 65.9 | 86.8 | 86.9 | 15.5 | 11.6 | 33.0 | 27.7 |
| NGCE (w/o $\mathcal{L}_C$) | 75.6 | 81.3 | 92.1 | 92.2 | 20.4 | 17.5 | 39.9 | 35.9 |
| NGCE (w/o $\mathcal{L}_{C1}$) | 79.4 | 83.9 | 93.7 | 93.7 | 20.7 | 17.5 | 40.2 | 36.4 |
| NGCE (w/o $\mathcal{L}_{C2}$) | 77.6 | 83.1 | 93.4 | 93.5 | 21.4 | 18.7 | 41.3 | 37.2 |
| NGCE (w/o $\widehat{\mathbf{A}}$) | 61.6 | 67.6 | 80.5 | 81.3 | 10.1 | 6.6 | 27.1 | 20.3 |
| NGCE (w/o $\mathbf{Z}$) | 68.3 | 71.4 | 89.8 | 90.9 | 17.1 | 15.6 | 35.1 | 31.3 |
| **NGCE** | **80.5** | **85.0** | **94.7** | **94.8** | **22.6** | **19.3** | **42.2** | **38.4** |

*Table 4.* The ablation study results of NGCE on ACM and Chameleon. The original results are shown in **bold**. w/o $\widehat{\mathbf{A}}$ denotes the replacement of joint process with adjacency matrix $\mathbf{A}$, w/o $\mathbf{Z}$ denotes the replacement of refined node features $\mathbf{Z}$ with an affinity matrix built directly form original node features.

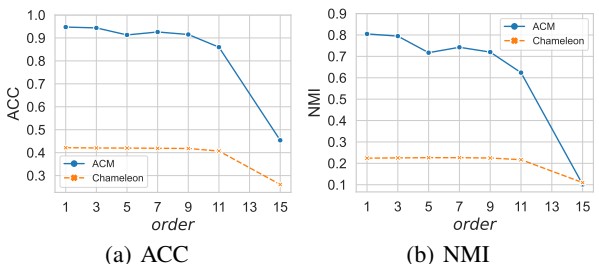

(a) ACC  (b) NMI

*Figure 2.* Parameter sensitivity analysis for *order* in homophilic graph dataset ACM and heterophilic graph dataset Chameleon.

## 4.2. Overall Results

Tables 2 and 3 present the results of all compared methods on homophilous and heterophilous graph datasets, from which we have the following observations. The experimental results on the homophilous graph dataset ACM demonstrate that NGCE is highly competitive with the state-of-the-art models, with significant improvements in all evaluative metrics. Moreover, NGCE outperforms other baselines on heterophilous graph datasets, *i.e.*, Chameleon and Wisconsin, which are typically challenging for most models. Our model performs better than the baselines in most of the evaluation metrics.

## 4.3. Ablation Studies and Analysis

### 4.3.1. Effect of each loss.

To understand the importance of the unsupervised node embeddings learning and contrastive learning, we removed reconstruction loss $\mathcal{L}_{Rec}$ of the autoencoder and the contrastive learning loss $\mathcal{L}_C$ for the proposed NGCE. Furthermore, for the contrastive learning loss $\mathcal{L}_C$, ablation study were also separately carried out on its components $\mathcal{L}_{C1}$ and $\mathcal{L}_{C2}$. We removed each loss in homophilic graph dataset ACM and heterophilic graph dataset Chameleon, respectively, to observe the changes in performance. The experimental results are shown in Table 4. As can be seen, $\mathcal{L}_{Rec}$

and $\mathcal{L}_C$ dominate the total losses in terms of their impact on model performance. To be more precise, the effects of $\mathcal{L}_{C1}$ and $\mathcal{L}_{C2}$ on the performance metrics of the model are remarkably similar. Implementing these components resulted in an increase in ACC of 1.0% and 1.3% on the ACM dataset and 2.0% and 1.0% on the Chameleon dataset. Furthermore, in comparison, $\mathcal{L}_{Rec}$ has a more significant impact on the performance metrics than $\mathcal{L}_C$, with the application of these losses leading to an increase in ACC of 7.9% and 2.6% on the ACM dataset, and 9.1% and 2.2% on the Chameleon dataset, respectively. Through ablation studies, we have validated the effectiveness of $\mathcal{L}_{C1}$ and $\mathcal{L}_{C2}$ within the contrastive learning loss $\mathcal{L}_C$. This confirms the efficacy of our dual contrastive learning designs: contrasting the noised node features processed by GCN with their original node features, and cross-view node-guided joint embeddings comparisons. These results demonstrate the effectiveness of including specific contrasting elements to improve the model's ability to generate strong and meaningful embeddings, utilizing both intra-view effectiveness and inter-view complementarity.

### 4.3.2. Effect of each component.

Since our model is based on the notion of node-guided graph encoding and consensus embedding encoding, the fusion of graph in joint process and node representation, along with the refinement of node features, is a critical component of NGCE. We conducted ablation studies to evaluate their impact on the performance of NGCE, where we replace graph joint encoded embeddings $\widehat{\mathbf{A}}$ with adjacency matrix $\mathbf{A}$, refined node features $\mathbf{Z}$ with affinity matrix based on original features $\mathbf{X}$, the results are detailed in Table 4. The results highlight that omitting either joint processing or node feature refinement significantly degrades model performance. Specifically, compared to the baseline model, the ablation experiments on the ACM dataset resulted in a decrease in ACC of 14.2% and 4.9%; on the Chameleon dataset, there was a decrease in ACC of 15.1% and 7.1%. Results high-

light the importance of refining node features, which leads to a meaningful representation in the node-guided process and validates the effectiveness of the strategy involving the weighted sum of the refined representational node similarity matrix with the original adjacency matrix.

### 4.3.3. PARAMETER SENSITIVITY ANALYSIS

The sensitivity analysis for $order$ hyperparameter in homophilic graph dataset ACM and heterophilic graph dataset Chameleon is shown in Fig. 2. From the spatial domain point of view, $order$ controls the aggregation order of the graph filter. The higher $order$ enables nodes to aggregate information from more distant ones, while nodes can only access feature information of closer nodes in lower orders. As can be seen from Fig. 2, ACC on the ACM dataset exhibits fluctuations around 93% when $order$ is less than 11, while on the Chameleon dataset ACC remains relatively stable at around 42% under the same condition. However, for $order$ values greater than 15, there is a sharp deterioration in the performance metrics for both datasets, dropping to around 45.4% for ACM and around 26.1% for Chameleon.

### 4.3.4. COMPLEXITY ANALYSIS

Let $M$ be the representation of the maximum number of neurons embedded within the hidden layers of the autoencoder, $Z$ denotes the maximum dimensionality of the embedding features, $T$ denotes the number of iterations for the outer loop, and $K$, $V$, and $N$ stand for the numbers of clusters, views, and examples respectively. The $K$-means and target distribution computations are performed once per outer loop iteration. Thus, their contribution to the total complexity will be $T \times O(NZK)$, giving us $O(TNZK)$. The autoencoder adheres to a complexity of $O(TNVM^2)$. The operation of GNN can be seen as multiplying an $N \times N$ matrix by an $N \times Z$ matrix, with a time complexity of $O(N^2Z)$ in an iteration. So, the total time complexity of our method is $O(TN(ZK + VM^2 + NZ))$. The complexity of NGCE can be further reduced by methods such as anchor graphs, but this is beyond the scope of the proposed method.

### 4.3.5. LIMITATION OF THE PROPOSED METHOD

Despite its demonstrated capabilities, NGCE depends on the availability of node features. This characteristic inherently limits its direct applicability to datasets where such features are sparse or absent. However, we posit that this limitation is not insurmountable and presents avenues for future investigation. To extend our approach to such scenarios, one potential strategy involves leveraging random walks to generate node embeddings. This technique, frequently employed in established graph embedding methods like DeepWalk (Perozzi et al., 2014), can capture structural information from the graph, thereby creating feature representations for nodes.

Another promising direction is developing and integrating a dedicated encoder that learns node features directly from the graph's topology. This would allow the model to function effectively even without pre-defined node attributes, significantly broadening its applicability. We plan to explore these extensions in future work.

## 5. Conclusion

In this paper, we introduce a novel graph-based contrastive learning framework for multi-view graph clustering, which effectively handles both homophily and heterophily in graph data. We propose that the homophilic and heterophilic information of graph data should not be separated, but rather processed within a unified framework to preserve its interactive essence. Consequently, we have developed a multi-view graph clustering method called Node-Guided Contrastive Encoding (NGCE). Specifically, NGCE includes a joint process that integrates a graph and node embedding similarity matrix sensitive to the degree of graph homophily, a contrastive learning-guided graph encoding mechanism driven by the degree of recovery of noise-enhanced node features, and a mechanism for contrastive fusion across views. The experimental results show that the proposed NGCE adeptly accommodates both homophilic and heterophilic datasets within the multi-view graph clustering domain, achieving state-of-the-art performance metrics.

## Acknowledgements

This work was supported in part by National Key Research and Development Program of China (Nos. 2024YFC2310800 and 2024YFC2310801), National Natural Science Foundation of China (No. 62476052), Sichuan Science and Technology Program (No. 2024NSFSC1473), and Central Guidance for Local Science and Technology Development Fund Projects (No. 2024ZYD0268).

## Impact Statement

This paper presents work whose goal is to advance the field of Machine Learning. There are many potential societal consequences of our work, none which we feel must be specifically highlighted here.

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

## A. Details of Experiments

### A.1. Metrics

Following previous works, four commonly used metrics, *i.e.*, normalized mutual information (NMI), adjusted rand index (ARI), accuracy (ACC), and F1 scores (F1), are adopted to evaluate the clustering performance.

### A.2. Visualization of learned consensus embedding

To unveil the inherent clustering structure, we have also employed $t$-SNE algorithm for the visualization of all six datasets to depict the distribution of the learned consensus embedding $\mathbf{H}$. As depicted in Fig. 3, the visual results highlight that NGCE exhibits an enhanced clustering structure.

### A.3. Details of Datasets

The experiments were conducted on six MVG datasets: ACM, DBLP, IMDB, Texas, Chameleon, and Wisconsin. Details and sources of the datasets are presented subsequently.

- Homophilous graph datasets: **ACM** is a paper network from the ACM database[1] and is composed of two graphs: the co-paper network and the co-subject network; **DBLP** is a network sourced from the DBLP database[2], consists of three graphs: co-author, co-conference, and co-term; **IMDB** is a movie network that originates from the IMDB dataset[3], including graphs of both co-actors and co-directors.
- Heterophilous graph datasets: **Texas** is a webpage graph from WebKB[4]; **Chameleon** is a subset of the Wikipedia network[5]; **Wisconsin** is webpage graph from WebKB[6]. Since Texas, Chameleon and Wisconsin are single-view graph data, we copy the graph as the second view.

## B. Algorithm

The proposed NGCE model is summarized in Algorithm 1. Algorithm 1 presents a comprehensive breakdown of the code flow.

## C. Preliminaries

In this section, we introduce a brief background of multi-view graph clustering and GCNs.

---

[1]https://dl.acm.org/

[2]https://dblp.uni-trier.de/

[3]https://www.imdb.com/

[4]http://www.cs.cmu.edu/afs/cs.cmu.edu/project/theo-11/www/wwkb

[5]https://github.com/benedekrozemberczki/MUSAE/

[6]http://www.cs.cmu.edu/afs/cs.cmu.edu/project/theo-11/www/wwkb

---

**Algorithm 1** Multi-View Graph Clustering via Node-Guided Contrastive Encoding (NGCE)

---

**Input:** The adjacency matrices $\{\mathbf{A}^n\}_{n=1}^V$ and node feature matrices $\{\mathbf{X}^n\}_{n=1}^V$

$Initialization\ phase$

Apply the $k$-means algorithm to original node feature matrices $\{\mathbf{X}^n\}_{n=1}^V$ for the pseudo label matrix $\bar{\mathbf{Y}}$ of the first iteration $Train\ phase$

**while** not reaching the maximum iterations or not reaching exit condition **do**

  **for** $n$ in $1, 2, \dots, V$ **do**

    Calculate $\mathbf{Z}^n$ and $\mathbf{X}^n_{\mathbf{pred}}$ based on Eq. (3) and (4)

    Calculate $\mathbf{S}^n$ based on Eq. (5)

    **If** it is the **first** iteration **do**

      Calculate $\boldsymbol{\omega}^n$ based on Eq. (6) with pseudo label matrix $\bar{\mathbf{Y}}^T$ from $Initialization\ phase$

    **else do**

      Calculate $\boldsymbol{\omega}^n$ based on Eq. (6) with pseudo label matrix $\bar{\mathbf{Y}}$ from last iteration

    Calculate $\widehat{\mathbf{A}}^n$ and $\widetilde{\mathbf{X}}^n$ based on Eq. (7) and (9)

    Calculate $\overline{\mathbf{X}}^n$ by $\overline{\mathbf{X}}^n = GCN(\widehat{\mathbf{A}}, \widetilde{\mathbf{X}})$ based on Eq. (10)

    Calculate $\mathbf{h}^n$ and $\boldsymbol{\omega}^n_h$ based on Eq. (13) and (16)

  **end for**

  Update $\mathbf{H}$ based on Eq. (15)

  **for** $v$ in $1, 2, \dots, V$ **do**

    Update $\bar{\mathbf{Y}}$ with $\mathbf{H}$ base on $k$-means

  **end for**

**end while**

**Output:** Consensus embedding $\mathbf{H}$

---

In the task of multi-view graph clustering, the objective is to group a set of $n$ nodes into $k$ clusters. To achieve this, we utilize $\mathcal{G} = (\mathcal{V}, \mathcal{E})$ to denote a graph. Here, $\mathcal{V}$ represents the nodes set, and the set of all nodes belonging to class $i$ is represented as $\mathcal{V}_i$, with $N = |\mathcal{V}|$, and $\mathcal{E} \subseteq \mathcal{V} \times \mathcal{V}$ represents the edge set with selfloops. The feature matrix for the nodes is denoted as $\mathbf{X} \in \mathbb{R}^{N \times d}$, and the symmetric adjacency matrix of the graph $\mathcal{G}$ is represented by $\mathbf{A} \in \mathbb{R}^{N \times N}$, with elements $a_{ij} = 1$ indicating the presence of an edge between node $i$ and node $j$, and $a_{ij} = 0$ otherwise. Additionally, we define the degree matrix of $\mathbf{A}$ as $\mathbf{D}_{ii} = \sum_j a_{ij}^n$, enabling the normalization of each view's $\mathbf{A}$ to $\widetilde{\mathbf{A}} = (\mathbf{D})^{-1}\mathbf{A}$. The normalized graph Laplacian matrix is defined as $\widetilde{\mathbf{L}} = \mathbf{I} - \widetilde{\mathbf{A}}$, with $\mathbf{I}$ representing the identity matrix.

The Graph Convolutional Networks (GCNs) leverages the principle of spectral graph theory to facilitate learning on graph-structured data. The core of this method is the utilization of the normalized graph Laplacian matrix to capture the graph's structural information. The GCNs operates by propagating node features across the graph's structure, thereby enabling each node to aggregate features from its neighbors,

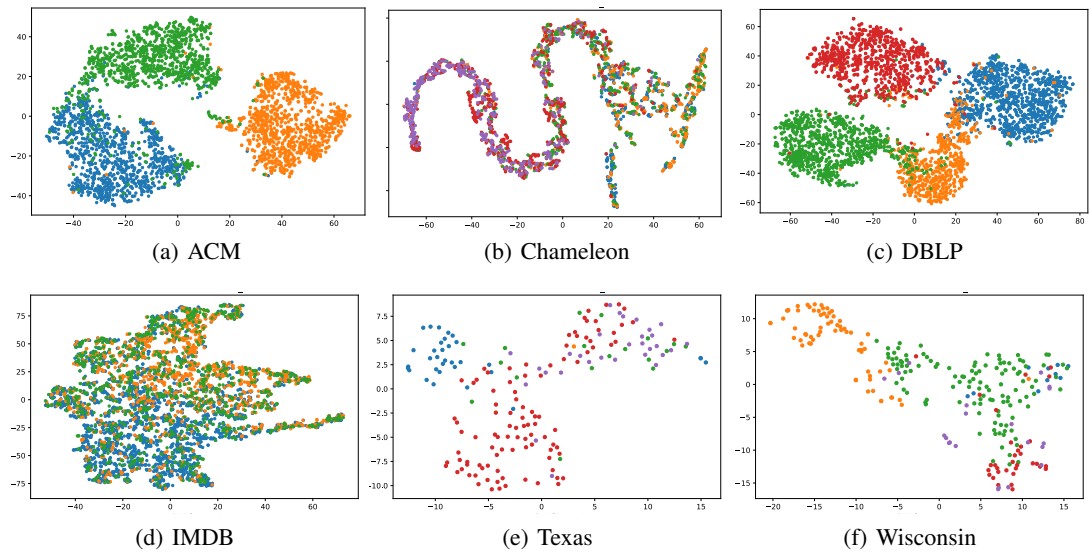

*Figure 3.* Visualization of learned consensus embedding **H** in all six datasets.

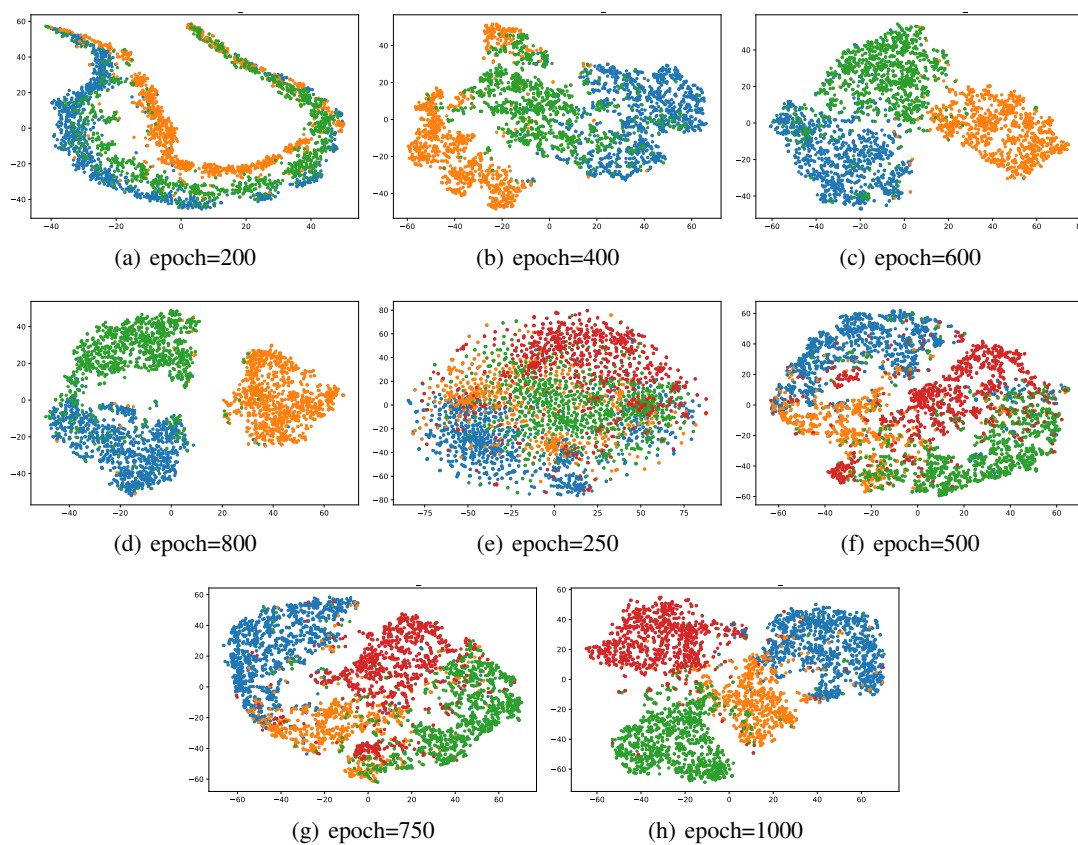

*Figure 4.* Visualization of learned consensus embedding **H** in the ACM dataset (Figure 4(a), 4(b), 4(c) and 4(d)) and DBLP dataset (Figure 4(e), 4(f), 4(g) and 4(h)) across epochs. At the start of the training stage, the embedded features are non-separable. As the training progresses, the clustering structures of the embedded features become more apparent while their centroids gradually separate.

effectively capturing both local and global graph structure.

In this paper, we use GCN without learnable parameters :

$$\mathbf{H}^{(l+1)} = \sigma\left(\mathbf{D}^{-\frac{1}{2}}\widetilde{\mathbf{A}}\mathbf{D}^{-\frac{1}{2}}\mathbf{H}^{(l)}\right), \qquad (16)$$

| Notations | Meaning |
|---|---|
| $\mathcal{G}$ | $\mathcal{G} = (\mathcal{V}, \mathcal{E})$, graph with node set $\mathcal{V}$ and edge set $\mathcal{E}$. |
| $\mathbf{X}$ | $\mathbf{X} \in \mathbb{R}^{N \times d}$, the feature matrix of the nodes. |
| $c$ | The number of clusters. |
| $\mathbf{A}$ | $\mathbf{A} \in \mathbb{R}^{N \times N}$, the adjacency matrix with self-loop. |
| $\mathbf{D}^{(n)}$ | Degree matrix in GCN. |
| $\mathbf{I}$ | Identity matrix. |
| $N$ | $N = |\mathcal{V}|$, the number of nodes. |
| $d$ | The dimension of original node features. |
| $V$ | The number of views. |
| $f^n(\cdot), g^n(\cdot)$ | The encoder and decoder in the autoencoders. |
| $d_l$ | Dimension of the distilled node embeddings. |
| $d_h$ | Hidden layer dimension of the autoencoder. |
| $n_l$ | The number of hidden layers. |
| $\mathbf{Z}^n$ | Distilled and refined node embeddings from $f^n(\cdot)$. |
| $\mathbf{X}^n_{\mathbf{pred}}$ | Output of the decoder $g^n(\cdot)$. |
| $\mathbf{S}^n$ | Node correlation matrix for the $n$-th view. |
| $\mathcal{S}(\cdot, \cdot), \mathrm{Sim}(\cdot, \cdot)$ | Cosine similarity. |
| $\boldsymbol{\omega}^n$ | Homophily wight for the $n$-th view. |
| $\bar{\mathbf{Y}}$ | $\bar{\mathbf{Y}} \in \{0,1\}^{N \times c}$, one-hot encoded pseudo labels. |
| $\widehat{\mathbf{A}}^n$ | Graph joint encoded embeddings for the $n$-th view. |
| $\mathbf{E}$ | $\mathbf{E} \in \{0,1\}^{N \times d}$, binary random masking matrix. |
| $p_c$ | Probability of Bernoulli distribution of random mask. |
| $\widetilde{\mathbf{X}}$ | Node features with random mask (noisy features). |
| $\overline{\mathbf{X}}^n$ | Recovered node features matrix for the $n$-th view. |
| $\mathbf{H}^{(l)}$ | Output of $l$-th layer GCN. |
| $order$ | Depth of GCN in NGCE. |
| $\mathbf{h}^n$ | Node-guided joint embeddings of the $n$-th view. |
| $\mathbb{1}_{[\cdot]}, \mathbb{I}^0_{ij}, \mathbb{I}^1_{ij}$ | Indicator function. |
| $\odot$ | Hadamard product. |
| $C_k$ | Cluster center for the $k$-th class of $\mathbf{H}$. |
| $c_i$ | The Class assigned to node $i$ in $\bar{\mathbf{Y}}$. |
| $\mathbf{H}$ | Consensus embedding. |
| $\rho$ | Cross-view weight modulation parameter |
| $\boldsymbol{\omega}^n_h$ | Cross-view fusion wight allocated to $\mathbf{h}^n$. |
| $eva^n$ | Function to measure congruence between $\mathbf{H}$ and $\mathbf{h}^n$. |
| $\mathcal{L}^n_{Rec}$ | Reconstruction loss on $\mathbf{X}^n_{\mathbf{pred}}$ and $\mathbf{X}$ of the $n$-th view. |
| $\mathcal{L}_{C1}, \mathcal{L}_{C2}$ | Contrastive learning loss. |
| $\eta$ | Learning rate. |
| $\lambda_{wd}$ | Weight decay regularization parameter. |

*Table 5.* Overall notations.

