# OpenReview forum: "Multi-View Graph Clustering via Node-Guided Contrastive Encoding"
_ICML.cc/2025/Conference — ICML 2025 poster_

### Official Review · Reviewer_WwHn · 2025-03-11

**Overall Recommendation:** 4

**Summary:**

This paper presented a novel approach to MVGC called Node-Guided Contrastive Encoding. This method addresses the challenges inherent in GNNs for clustering by effectively using homophilic and heterophilic information within graph data. The proposed framework uses node features to guide the embedding process, thus preserving the interactive nature of graph information.

**Claims And Evidence:**

Yes

**Essential References Not Discussed:**

The paper comprehensively reviews the most relevant literature in the fields of Multi-View Clustering and Graph Learning.

**Experimental Designs Or Analyses:**

Yes. The experimental setting and results have been reviewed.

**Methods And Evaluation Criteria:**

Yes

**Other Comments Or Suggestions:**

There are some typos, such as the explanation below Equation (2), and a repeated definition of the Hadamard product.

**Other Strengths And Weaknesses:**

Strengths

1. The NGCE framework introduces a unique way of integrating node features with graph structure through contrastive learning that combines homophilic and heterophilic information.

2. The use of noise-augmented node feature recovery and adaptive weighting mechanisms demonstrates a well-considered design aimed at improving MVG learning.

Weaknesses

1.The time complexity of the proposed NGCE model may raise concerns, as it appears to generate V views iteratively.

2.Some aspects of the proposed method may appear incremental, particularly the use of graph neural networks in a contrastive learning setting, which has been explored in prior works.

3.The performance of NGCE in purely structural graph clustering scenarios, where node features are absent, remains unexplored.

**Questions For Authors:**

Note that node-guided encoding in NGCE exhibits strong dependency on node features, could the methodology remain functional when applied to pure graph-structured datasets lacking node features?

**Relation To Broader Scientific Literature:**

The key contributions of the paper are related to the broader scientific literature on Multi-View Clustering and Graph Learning.

**Theoretical Claims:**

Yes. The methodology of the paper has been reviewed.

---

> ### Author Rebuttal · Authors · 2025-04-01
>
> Thank you for your insightful feedback and constructive critiques on our manuscript. Below, we provide a point-by-point response to your comments.
>
> **Q1: Time Complexity Concerns**
> **A1:** We appreciate the reviewer’s attention to computational efficiency. While generating *V* views iteratively introduces additional complexity, this design aligns with state-of-the-art late-fusion multi-view clustering methods (e.g., DualGR), where view-specific processing precedes fusion. As detailed in Section 4.3.4 (Complexity Analysis), NGCE’s complexity remains comparable to prior works, as the incremental cost arises solely from a parallel GNN module (for node-guided encoding, the part from $\tilde{X}$  to  $\overline{X}^n$ in Figure 1).
>
> **Q2: Incremental Contribution**
> **A2:** While NGCE builds on contrastive learning principles, its *node-guided encoding* mechanism represents a novel contribution by unifying homophilic and heterophilic interactions within a single framework. Unlike prior works that treat these as separate tasks or rely on heuristic aggregation, NGCE explicitly models their interplay through adaptive weight learning and noise-augmented recovery (Section 3.1.2). This design enables superior performance on both homogeneous and heterogeneous graphs.
>
> **Q3: Applicability to Purely Structural Graph and Functionality Without Node Features**
>
> **A3:** We fully acknowledge your concern regarding the method's reliance on node features, which indeed limits  its applicability in datasets with limited or no node feature  information. In the revised *Limitations* subsection in Discussion section, we clarify that NGCE’s current implementation assumes the availability of node features.  Specifically, we will acknowledge that the current  implementation struggles in scenarios where node features are absent.  However, we believe that this limitation is not insurmountable. There  are potential solutions to extend our approach to such datasets, and we  intend to explore them in future work. Possible strategies include:
>
> - Random Walk-based Feature Generation: Random Walks could be employed to generate node embeddings by capturing structural information from  the graph, an approach often used in graph embedding techniques such as  DeepWalk.
> - Learned Node Features: We could also incorporate a dedicated encoder to generate node features directly from the graph structure, thereby enabling the method to function even in the absence of explicit node  attributes.
>
> Both strategies might allow us to retain the benefits of our  node-guided contrastive encoding approach while extending its  applicability to a wider range of datasets. We will present these  potential solutions in the *Limitations* subsection to provide a more comprehensive view of possible extensions to our work.
>
> **Q4: Typos and Redundant Definitions**
> **A4:** In response, we have carefully reviewed the entire manuscript and addressed all identified issues to improve clarity, consistency, and overall presentation.
>
> We sincerely appreciate your thorough review and constructive feedback, which have greatly improved the quality of our manuscript.

---

> > ### Comment · Reviewer_WwHn · 2025-04-07
> >
> > Thank you for the author's reply and clarification. The author's reply resolved my doubts and questions, so I decided to change the rating to accept.

---

> > > ### Author Response · Authors · 2025-04-08
> > >
> > > Thank you for your time and constructive feedback. We appreciate your consideration of our revisions.

---

### Official Review · Reviewer_KgQm · 2025-03-13

**Overall Recommendation:** 4

**Summary:**

This paper introduces Node-Guided Contrastive Encoding (NGCE), a novel framework for multi-view graph clustering that integrates homophilic and heterophilic information through node-guided contrastive learning. NGCE aims to outperform existing methods by emphasizing node feature-based information and avoiding explicit decoupling of homophilic and heterophilic components. Experimental results on six benchmark datasets demonstrate significant improvements over baselines.

**Claims And Evidence:**

The claims made are clearly explained in the manuscripts.

**Essential References Not Discussed:**

No.

**Experimental Designs Or Analyses:**

The experimental design is relatively reasonable. The experimental results outperform almost all baseline methods, and the experimental analysis is also relatively complete.

**Methods And Evaluation Criteria:**

Yes, the proposed method provides a novel solution in this field.

**Other Comments Or Suggestions:**

Please refer to the strengths and weaknesses.

**Other Strengths And Weaknesses:**

Strengths:
1. To advance existing multi-view graph contrastive learning approaches which primarily focus on homophilic graphs, NGCE introduces node-guided encoding to unify homophilic and heterophilic interactions.
2. NGCE can improve compatibility with GNN filtering, effectively integrate both homophilic and heterophilic information, and enhance contrastive learning across multiple views.
3. The paper is well organized and sufficient experiments have been conducted.

Weaknesses:
1. The paper delays defining mathematical symbols until the appendix, rather than introducing them upon their first appearance, which disrupts readability and comprehension.
2. Certain equations, such as Eq. 11, are not clearly presented, potentially causing misunderstandings about the computational process. Is that GCN in Eq. 11 set the same as that in Eq. 8?

**Questions For Authors:**

Please refer to the weaknesses.

**Relation To Broader Scientific Literature:**

The proposed NGCE framework introduces novel methodologies to the domains of graph machine learning, multi-view graph contrastive learning, and the handling of heterophily in GNNs.

**Theoretical Claims:**

The theory in the manuscript provides further clarification and explanation of the proposed methodology.

---

> ### Author Rebuttal · Authors · 2025-04-01
>
> Thank you for your insightful feedback and constructive critiques on our manuscript. Below, we provide a point-by-point response to your comments.
>
> **Q1: Delayed Definition of Mathematical Symbols**
> **A1:** We sincerely apologize for the oversight in deferring symbol definitions to the appendix. In the revised manuscript, all mathematical symbols (e.g., adjacency matrices, node features, and operators) are now explicitly defined when they first appear in the main text. For example, all parameters related to GNNs are now clearly introduced within their corresponding equations.
>
> **Q2: Ambiguity in Equation (11)**
> **A2:** We thank the reviewer for identifying this ambiguity. The GCN architecture in Equation (11) shares the *same structural design* as the GCN in Equation (8) and operates with *independent parameters* (i.e., no weight sharing). This distinction ensures that the node recovery module learns distinct patterns from the graph encoding process. We have clarified this in the revised manuscript by explicitly stating that in Equation (11).
>
> We greatly appreciate your careful evaluation and thoughtful recommendations, which have significantly improved the rigor and clarity of our work.

---

### Official Review · Reviewer_nDQ8 · 2025-03-14

**Overall Recommendation:** 3

**Summary:**

This work primarily focuses on integrating homogeneous and heterogeneous information in graph data into a unified framework. Its core modules are an edge and node embedding similarity matrix sensitive to graph homophily, a contrastive learning-guided graph encoding mechanism driven by the recovery of noise-enhanced node features, and a contrastive fusion mechanism across views. The most innovative contribution is the introduction of the second module, which enables the simultaneous learning of homogeneous and heterogeneous information. Extensive experiments demonstrate the feasibility and effectiveness of the model.

**Claims And Evidence:**

Yes, the claims are made in the submission supported by clear and convincing evidence.

**Essential References Not Discussed:**

N/A.

**Experimental Designs Or Analyses:**

Yes, the experimental designs and analyses are reasonable.

**Methods And Evaluation Criteria:**

Yes, the proposed methods make sense for the problem.

**Other Comments Or Suggestions:**

1. In Eq. (9), the statement "the first term minimizes the agreement between the node and its non-neighbors…" appears to be grammatically incorrect. What is more, its intended meaning seems problematic.
2. In Figure 2, the clustering performance is consistently best when the order is 1. The authors should further explain this particular phenomenon.
3. The comparison algorithms lack references to works published in 2024.

**Other Strengths And Weaknesses:**

Strengths:
1. The primary motivation of this work is to integrate the learning of heterogeneous and homogeneous graphs within a unified framework. Its innovative intent is distinctly evident.
2. The work exhibits a clear logical structure.


Weaknesses:
1. In Section 3.1.2, the recovery of noise-enhanced node features is claimed to facilitate the learning of heterogeneous information. However, this lacks theoretical justification. Furthermore, superior clustering performance on heterogeneous graphs in experiments does not necessarily imply that this improvement is attributed to the recovery of noise-enhanced node features. Therefore, the authors should provide further theoretical validation or more detailed experimental evidence.
2. The manuscript contains numerous typographical errors that need to be corrected.

**Questions For Authors:**

See weakness and suggestions.

**Relation To Broader Scientific Literature:**

Existing MVGC models typically handle heterogeneous and homogeneous graphs separately. The motivation of this work is to integrate heterogeneous and homogeneous graphs into a unified framework. This motivation is innovative.

**Theoretical Claims:**

Yes, the proofs for theoretical claims are correct.

---

> ### Author Rebuttal · Authors · 2025-04-01
>
> Thank you for your insightful feedback and constructive critiques on our manuscript. Below, we provide a point-by-point response to your comments.
>
> **Q1: Theoretical Validation or Detailed Experimental Evidence of Section 3.1.2**
>
> **A1:** Thank you for your valuable feedback. In the current version, we have provided a detailed algorithmic description and conducted extensive ablation studies to empirically validate the effectiveness of our proposed mechanism. We fully acknowledge the importance of theoretical analysis and plan to incorporate rigorous theoretical examinations (including convergence analysis and generalization capability analysis) in future extensions of this work.
>
> As shown in the following tables, removing the node recovery component ("NGCE w/o node recovery") leads to significant performance degradation across both homogeneous and heterogeneous graphs.
>
> |Methods|ACM|DBLP|IMDB|texas|Chameleon|Wisconsin|
> |--|--|--|--|--|--|--|
> |NGCE (w/o node recovery)|75.0/79.7/92.8/92.8|62.1/62.4/84.8/85.2|3.5/6.7/48.5/40.5|28.4/25.0/55.2/34.9|19.9/15.3/40.8/35.7|36.7/30.5/58.6/43.3|
> |NGCE|80.5/85.0/94.7/94.8|79.1/84.0/93.3/92.8|5.6/12.7/54.6/43.4|47.8/54.9/77.6/46.2|22.6/19.3/42.2/38.4|46.8/46.4/73.7/47.2|
>
> This table reports clustering performance metrics in the order of NMI/ARI/ACC/F1. These results suggest that the recovery mechanism plays a critical role in capturing both local and global structural patterns, especially in heterogeneous datasets.
>
> **Q2: Typographical Revisions**
>
> **A2:** In response, we have carefully reviewed the entire manuscript and addressed all identified issues to improve clarity, consistency, and overall presentation.
>
> **Q3: Equation (9) Clarification**
>
> **A3:** We have revised the explanation surrounding this equation to provide clearer context and interpretation. Specifically, the first term minimizes the agreement between the reconstructed node features and the features of the masked nodes in the encoder graph, since there is no valid information correlation in these nodes; the second term drives the GCN with graph joint encoding embeddings to retain critical unmasked node information.
>
> The whole process requires the GCN to identify and preserve the correct unmasked features from the masked non-adjacent nodes in the graph structure, where the graph joint encoding embeddings are forced to learn the essential patterns of node feature distribution in the encoding stage.
>
> **Q4: First-Order Neighborhood Superiority (Figure 2)**
>
> **A4:** We believe that the superior performance under first-order neighborhood aggregation is due to the graph encoding mechanism proposed in our framework. This phenomenon suggests that our learning paradigm enables the encoded graph to directly integrate valuable edges, thereby substantially reducing the necessity for higher-order neighborhood processing in subsequent GNN operations. Specifically, when the order is set to 1 or some other low number, the model effectively captures direct neighborhood relationships within the encoded graph while minimizing noise interference. In contrast, although higher-order aggregation introduces redundancy and can theoretically provide additional information, the induced noise becomes particularly difficult to mitigate in unsupervised learning scenarios due to the lack of explicit supervisory signals. A discussion of this trade-off, along with the inherent limitations of our model, will be provided in the Discussion section.
>
> **Q5: 2024 Baseline Comparisons.**
>
> **A5:** We added comparisons with VGMGC [TNNLS 25], BMGC [ACM MM 24], and SMVC [Neural Networks 24]. Among these, VGMGC and BMGC are open-source and can be evaluated across all datasets, while SMVC, though not yet open-sourced, shares our homogeneous dataset selection. The comparison results are shown in the table below. For open-source baselines, we used the results reported in the original paper where available; otherwise, we evaluated them using default parameters or settings from similar datasets. The following table reports clustering performance metrics in the order of NMI/ARI/ACC/F1.
>
> |Methods|ACM|DBLP|IMDB|Texas|Chameleon|Wisconsin|
> |--|--|--|--|--|--|--|
> |SMVC|72.4/78.0/92.3/92.0|76.1/81.6/92.4/92.0|8.0/7.2/41.3/37.2|-/-/-/-|-/-/-/-|-/-/-/-|
> |BMGC|78.4/83.3/94.1/94.2|80.1/85.4/94.0/93.6|5.5/4.9/44.1/40.7|29.1/15.8/42.5/38.3|9.4/5.9/30.8/30.7|34.0/24.3/51.5/40.8|
> |VGMGC|76.3/81.9/93.6/93.6|78.3/83.7/93.2/92.7|0.8/3.2/52.6/32.8|35.4/26.0/55.2/46.9|22.4/13.4/40.1/39.5|41.6/34.8/56.6/49.6|
> |NGCE (ours)|80.5/85.0/94.7/94.8|79.1/84.0/93.3/92.8|5.6/12.7/54.6/43.4|47.8/54.9/77.6/46.2|22.6/19.3/42.2/38.4|46.8/46.4/73.7/47.2|
>
> These experimental results show that our method consistently achieves superior performance compared to methods in recent publications.
>
> We greatly appreciate your rigorous evaluation and suggestions, which have significantly strengthened our manuscript.

---

> > ### Comment · Reviewer_nDQ8 · 2025-04-08
> >
> > Thank you for the author's reply. I read the author's response. Combining the overall manuscript and the responses, I raise my score.

---

> > > ### Author Response · Authors · 2025-04-08
> > >
> > > We are grateful for your comments and the opportunity to improve our work. Thank you for your updated assessment.

---

### Decision · Program_Chairs · 2025-05-01

**Decision:**

Accept (poster)

**Comment:**

All the three reviewers agree that the paper is interesting, the proposed method has made important contribution to the analysis of the homogeneous and heterogeneous graphs and demonstrated better performance in different metrics compared to the existing ones. The rebuttal made various clarifications and explanations on the concerns/comments raised and provided more experimental results.